# Design and evaluation of an MRI-ready, self-propelled needle for prostate interventions

**Jette Bloemberg**[1]*, **Fabian Trauzettel**[1], **Bram Coolen**[2], **Dimitra Dodou**[1], **Paul Breedveld**[1]

1 Bio-Inspired Technology Group (BITE), Department of Biomechanical Engineering, Faculty of Mechanical, Maritime and Materials Engineering, Delft University of Technology, Delft, The Netherlands, 2 Department of Biomedical Engineering & Physics, Amsterdam University Medical Centers (AUMC), Amsterdam, The Netherlands

* J.Bloemberg@tudelft.nl

**Data Availability Statement:** All relevant data are within the paper and its Supporting information files.

## Abstract

Prostate cancer diagnosis and focal laser ablation treatment both require the insertion of a needle for biopsy and optical fibre positioning. Needle insertion in soft tissues may cause tissue motion and deformation, which can, in turn, result in tissue damage and needle positioning errors. In this study, we present a prototype system making use of a wasp-inspired (bioinspired) self-propelled needle, which is able to move forward with zero external push force, thereby avoiding large tissue motion and deformation. Additionally, the actuation system solely consists of 3D printed parts and is therefore safe to use inside a magnetic resonance imaging (MRI) system. The needle consists of six parallel 0.25-mm diameter Nitinol rods driven by the actuation system. In the prototype, the self-propelled motion is achieved by advancing one needle segment while retracting the others. The advancing needle segment has to overcome a cutting and friction force while the retracting needle segments experience a friction force in the opposite direction. The needle self-propels through the tissue when the friction force of the five retracting needle segments overcomes the sum of the friction and cutting forces of the advancing needle segment. We tested the performance of the prototype in *ex vivo* human prostate tissue inside a preclinical MRI system in terms of the slip ratio of the needle with respect to the prostate tissue. The results showed that the needle was visible in MR images and that the needle was able to self-propel through the tissue with a slip ratio in the range of 0.78–0.95. The prototype is a step toward self-propelled needles for MRI-guided transperineal laser ablation as a method to treat prostate cancer.

## 1. Introduction

### 1.1 Focal laser ablation

Prostate cancer is the second most common cancer diagnosed in men and the fifth leading cause of cancer-related deaths for men worldwide in 2020 [1]. When prostate cancer is diagnosed at an early stage, it can be treated locally using focal therapy that reduces the risk of side effects by preserving noncancerous tissue [2]. Focal laser ablation of the prostate is an

**Funding:** This work was supported by the Netherlands Organisation for Scientific Research (Nederlandse Organisatie voor Wetenschappelijk Onderzoek, NWO), domain Applied and Engineering Sciences (TTW), and which is partly funded by the Ministry of Economic Affairs. Grant number 80450, Perspectief programme, Photonics Translational Research – Medical Photonics (MEDPHOT), awarded to PB. URL: https://www.nwo.nl/. Fabian Trauzettel is part of the ATLAS project. This project has received funding from the European Union's Horizon 2020 research and innovation programme under the Marie Sklodowska-Curie, grant number 813782. URL: https://atlas-itn.eu. The funders had no role in study design, data collection and analysis, decision to publish, or preparation of the manuscript.

**Competing interests:** The authors have declared that no competing interests exist.

appealing focal therapy option as it leads to homogeneous tissue necrosis caused by a laser fibre and does not appear to alter the sexual and urinary function of the patient [3].

Prostate cancer diagnosis and focal laser ablation require needle insertion to obtain core biopsies [4, 5] and position optical fibres near the target zone [6]. To this end, the clinician inserts the needle by pushing it through the tissue, which might lead to tissue strain in the needle vicinity [7], which in turn might cause functional damage to the surrounding tissues and organs [8], including the urethra, the rectum's anterior wall, and the pelvic sidewall [9]. Moreover, tissue motion and deformation might lead to needle positioning errors and poor control of the needle path [10]. As a result, clinicians typically need multiple attempts to reach the target location, leading to an increased risk of tissue damage [7]. Moreover, pushing the needle through the tissue requires an axial force on the needle. When this axial force exceeds the needle's critical load, the needle will deflect laterally—a phenomenon called buckling [11]. The lateral deflection might damage tissue in the needle vicinity and lead to poor control of the needle path [12, 13].

## 1.2 State-of-the-art in self-propelled needles

In an attempt to reduce tissue damage during needle insertion, needle designs have been developed that can be advanced without being pushed through the tissue. For instance, Ilami *et al.* [14] developed a needle with a magnetic tip that utilizes electromagnetic force and torque actuation to advance the needle through the tissue. Schwehr *et al.* [15] proposed a needle design that likewise utilizes electromagnetic torque to steer combined with a screw tip to allow the needle to pull itself through the tissue. A disadvantage of needle designs that utilize an electromagnetic field is that they are not compatible with magnetic resonance imaging (MRI). Besides electromagnetically actuated needles, wasp-inspired self-propelled needles have been developed [7, 16–18]. Female parasitic wasps pass their eggs through an ovipositor into their hosts, which sometimes hide in a solid substrate such as wood [19]. The tube-like ovipositor consists of three slender, parallel-positioned segments, called valves [20], which advance and retract with respect to each other in a reciprocating manner [20] (Fig 1A). A groove-and-tongue mechanism interlocks the valves along their length [21, 22]. The advancing-retracting motion of the valves has two functions. First, it keeps the unsupported length of the individual valves low [11]. Second, moving the individual valves forward one by one while retracting the others provides stability to the wasp's ovipositor and prevents buckling [11, 23]. The advancement and retraction forces produce a net force near zero, enabling a self-propelled motion.

Self-propelled needles do not require an external push force to advance through the tissue. They consist of multiple parallel segments that can slide along each other. The self-propelled motion is achieved by counterbalancing the cutting and friction force of the advancing segments by the friction force generated by other stationary or retracting segments [18]. For a self-propelled motion of the needle, Eq 1 holds:

$$\sum_{i=1}^{p} (\mathbf{F}_{\text{fric},i} + \mathbf{F}_{\text{cut},i}) \leq \sum_{j=1}^{r} (\mathbf{F}_{\text{fric},j}) \tag{1}$$

where $p$ is the number of advancing needle segments, $r$ is the number of retracting needle segments, and $\mathbf{F}_{\text{fric}}$ and $\mathbf{F}_{\text{cut}}$ are the friction and cutting force, respectively (Fig 1B). For the self-propelled motion to occur, the friction force of the retracting needle segments should overcome the sum of the friction and cutting forces of the advancing needle segments. In this way, the needle as a whole self-propels through the tissue by gradually moving the needle segments forward.

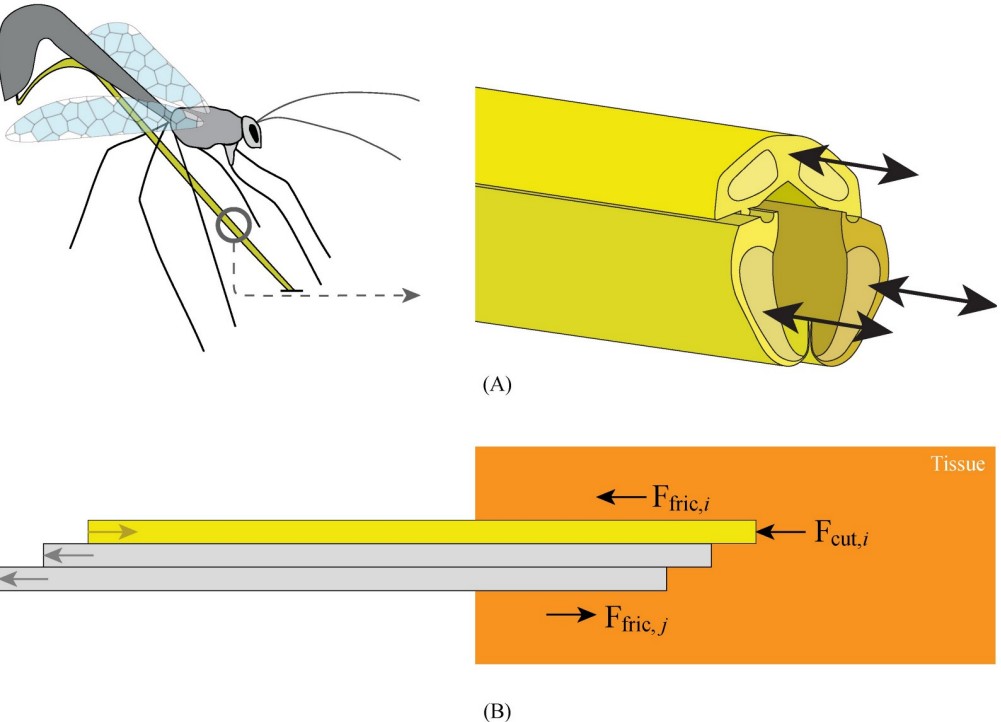

**Fig 1. Visualisation of the motion sequence of the ovipositor of a parasitic wasp.** (A) The ovipositor consists of three parallel valves that can move reciprocally (based on Cerkvenik *et al.* [20]). (B) Schematic illustration of ovipositor-inspired needle insertion into tissue with one advancing needle segment (yellow) and two retracting needle segments (grey). $F_{fric,i}$ is the friction force along the advancing needle segment, $F_{cut,i}$ is the cutting force on the tip of the advancing needle segment, and $F_{fric,j}$ is the friction of the retracting needle segments, which works in the opposite direction as the friction force of the advancing needle segments.

Worldwide, a number of ovipositor-inspired needles have been developed so far. Oldfield *et al.* [24], Frasson *et al.* [25], and Leibinger *et al.* [7] showed that tissue motion and damage around a needle are reduced when using a multi-segmented needle actuated with a reciprocal advancing-retracting motion compared to pushing the needle as a whole through the tissue. Parittotokkaporn *et al.* [23] showed that probes with a directional friction pattern inspired by the wasp ovipositor, actuated with an advancing-retracting motion, could move tissue along the needle surface without applying an external push force to the tissue. Scali *et al.* [26, 27] replaced the complex-shaped interlocking groove-and-tongue mechanism of the wasp valves with Nitinol rods devoid of serrations and bundled by a shrinking tube, resulting in an ultra-thin 0.4-mm diameter needle with six longitudinal segments [27]. Actuated by electric motors, the needle self-propels without buckling by advancing one needle segment at a time and slowly retracting the other five segments [26]. Furthermore, it is possible to steer the needle by inducing an offset between the needle segments, creating a discrete bevel-shaped tip [26].

### 1.3 Goal of this research

Wasp-inspired self-propelled needles could enable the clinician to reach the target tissue while avoiding unwanted tissue damage in and around the prostate. To guide needle positioning for focal laser ablation, MRI is an attractive imaging option because it provides visualisation of the target zone and real-time temperature monitoring [28, 29]. Current prototypes of wasp-inspired self-propelled needles use electric motors to actuate the individual needle segments

[18, 25]. These needles cannot be used in MRI-guided procedures, as the electric motors interfere with the magnetic field. The aim of this research was, therefore, to design an MR-safe actuation system for a self-propelled needle and to evaluate its performance in human prostate tissue.

## 2. Design

### 2.1 Design requirements

The complete design, called Ovipositor MRI-Needle, consists of a needle and an actuation unit. Following the design of Scali *et al.* [27], we decided to focus our research on a self-propelled wasp-inspired needle consisting of *six parallel needle segments* with a central lumen. To reach the prostate transperineally [30], we opted for a *needle length* of 200 mm. To comply with conventional needles used for optical biopsy and optical treatment fibre positioning [6], we used a *maximum needle diameter* of 1 mm. To enable evaluation in a closed-bore preclinical 7-T MRI system (MR Solutions, Guildford, United Kingdom) with an inner diameter of the radiofrequency (RF) coil of 65 mm, we developed an actuation unit fitting within this coil, the *actuation unit's diameter* not exceeding 65 mm, and the actuation unit containing a 2-mm diameter *central hollow core* to allow insertion of a functional element, such as an optical fibre. Finally, the *materials* used in the needle and the actuation unit are MRI-compatible to allow placing them inside the MRI system.

### 2.2 Overall system design

The needle's self-propelled motion requires a sequential translation of the six needle segments in six steps per cycle. During every step of the motion, one needle segment moves forward over a specified distance called the "stroke," while the other five needle segments move slowly backwards over one-fifth of the stroke distance (Fig 2A). The needle segments are continuously in motion in order to apply a constant strain to the surrounding tissue. We opted for a manually-controlled actuation system that allows the operator to drive the needle in simple discrete actuation steps, avoiding the need to set the exact advancing or retracting distance for each needle segment during each actuation step. Fig 2B shows how the operator drives the actuation unit by a stepwise manual translation of a translation ring (in red). The actuation unit converts the reciprocating motion of the translation ring into a global rotating motion of an internal selector, after which the selector selects and actuates the needle segments in the required order and over the required distance.

**Selector.** The design of the selector is based on the so-called click-pen mechanism of a ballpoint pen, Fig 3 [31]. The click-pen-mechanism converts the discrete motion of pressing the button at the end of the pen into a rotation and a subsequent translation of the ballpoint tip. Fig 4 shows the working principle of our selector (in green). The cylindrical mechanism is simplified and visualised in a two-dimensional (2D) schematic illustration to explain the working principle. The columns in Fig 4 show the subsequent steps in the motion cycle. The rows in Fig 4 show the different layers of the selector. The selector (Fig 4A–4D) contains two sets of aligned teeth. A fixed housing (in grey) also contains two sets of teeth. For the housing, the teeth at the right are shifted over half a tooth width. The selector is actuated by the input motion: a reciprocating translating motion in the horizontal x-direction. When the operator moves the selector in the positive x-direction (Fig 4A), the teeth on the right side of the selector come in contact with the teeth on the right side of the housing (Fig 4B, the interacting teeth of the housing are indicated in dark grey). The interaction between the teeth causes the selector to move in the negative y-direction over half the pitch distance of the teeth until the selector motion is blocked by the teeth so that it cannot move any further. In the following step, the

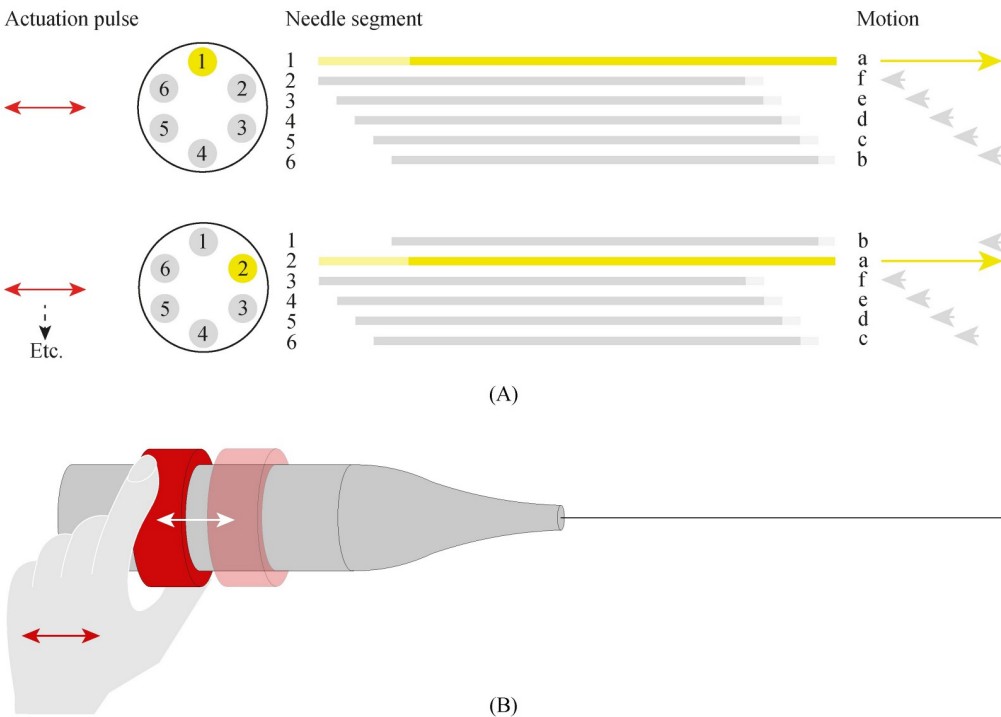

**Fig 2. Visualisation of the motion sequence of the needle segments.** (A) During the motion, one needle segment moves forward over the stroke distance while the other needle segments move slowly backwards over one-fifth of the stroke distance in a consecutive manner. (B) Manual translation of a translation ring (red) drives the actuation system. The actuation system converts the reciprocating motion of the translation ring into a sequential translation of the six needle segments.

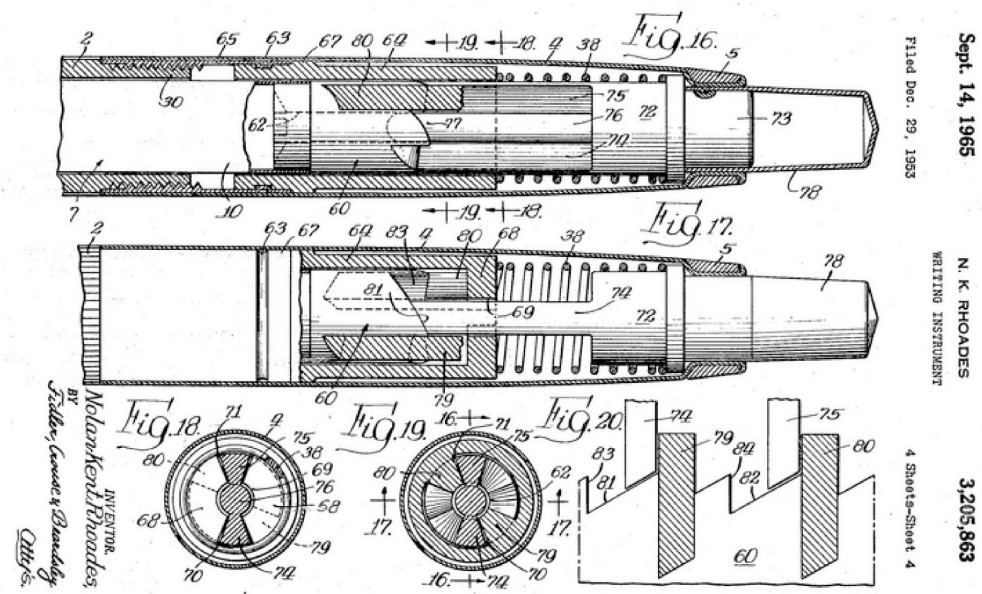

**Fig 3. Click-pen mechanism of a ballpoint pen.** Illustration of one of the first patented click-pen mechanisms (Parker Pen Co Ltd) [31].

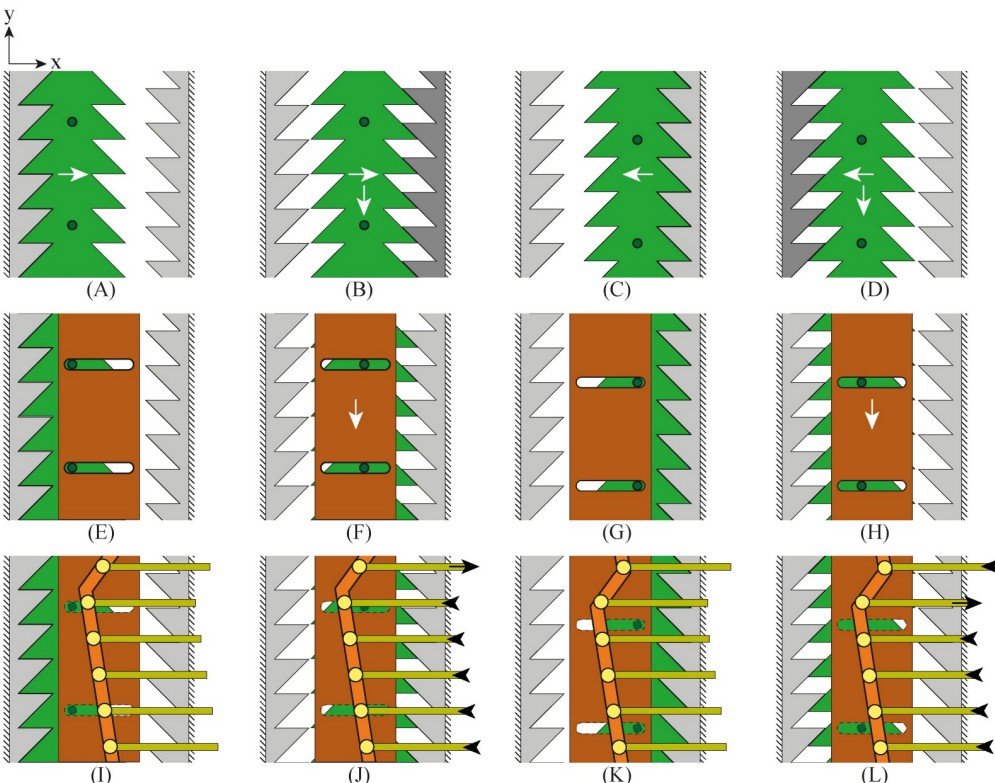

**Fig 4. Schematic representation of the selector motion mechanism in 2D, including the selector (green), housing (grey), cam (orange), and needle segment holders (yellow).** The columns show the subsequent steps in the motion cycle. The rows show the different layers of the selector.

selector is moved in the negative x-direction (Fig 4C) until the selector's left teeth come in contact with the housing's left teeth (Fig 4D), causing the selector to move again in the negative y-direction over half the teeth' pitch distance until the motion is again blocked. The interaction between the teeth of the selector and the housing converts the reciprocating horizontal motion that actuates the selector into a stepwise vertical translation.

**Cam.** Fig 4E–4H show how the selector contains small protruding cylinders (in dark green) that can slide in straight horizontal slots in a cam (in orange). The housing prevents the cam from translating in the horizontal x-direction. When the selector is translated in the positive or negative x-direction, the protruding cylinders transmit the selector's stepwise translation in the y-direction to the cam. Fig 4I–4L show that the cam contains a V-shaped slot (in light orange). Six needle segment holders (Fig 4I–4L, in yellow) contain small protruding cylinders (in light yellow) that can slide in the cam's V-shaped slot. The housing restricts the motion of the needle segment holders to a translation in the x-direction driven by the motion of the V-shaped slot. The asymmetric shape of the V causes one needle segment holder to move in the positive x-direction, with the other needle segment holders moving slowly in the negative x-direction.

**Working principle in 3D.** The stepwise translation of the selector in the y-direction in the simplified 2D illustration in Fig 4 is, in reality, a stepwise rotation around the x-axis in 3D. Fig 5 shows the 3D working principle of the selector (in green), surrounded by a concentric housing (in grey) and driving the six needle segment holders (in yellow) via the cam (in orange). The inside of the mechanism contains a hollow core to introduce additional instrumentation.

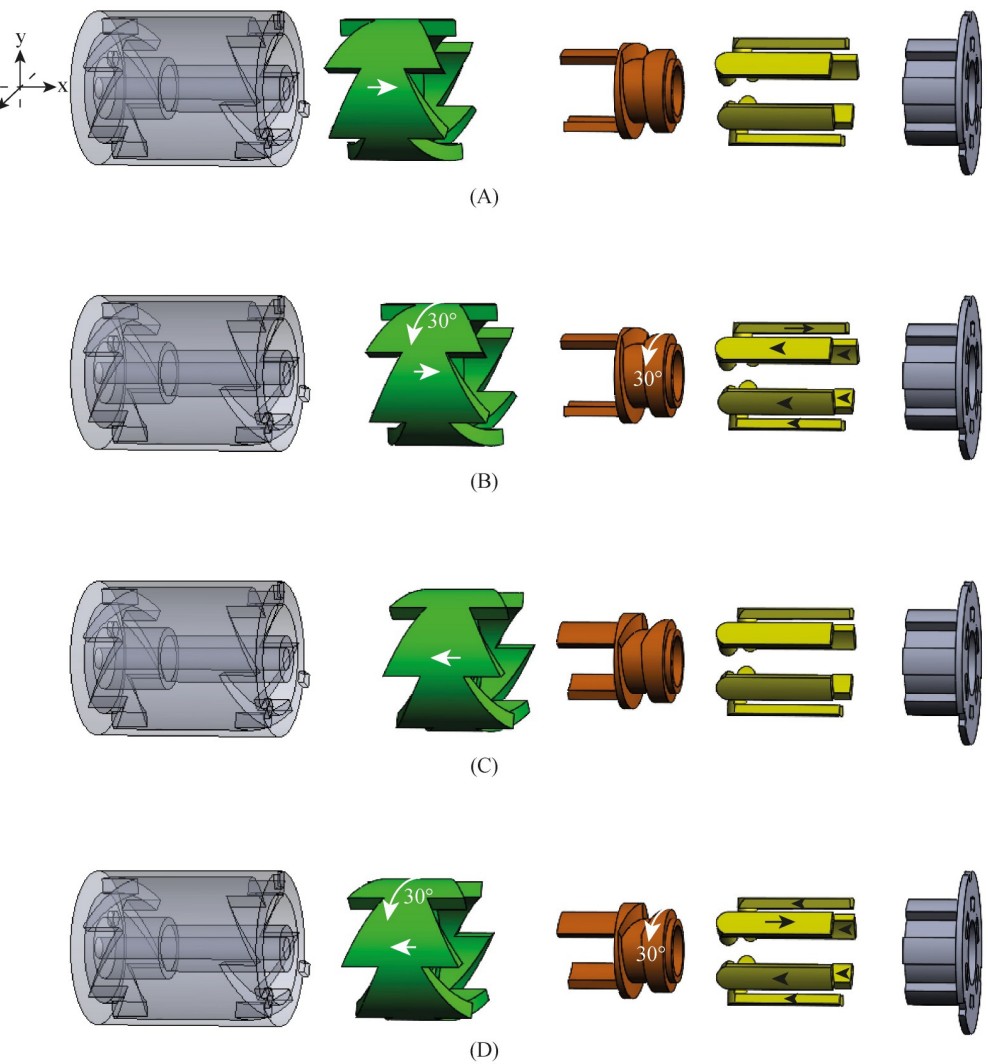

**Fig 5. Schematic motion sequence of the selector in 3D, including the selector (green), housing (grey), cam (orange), and needle segment holders (yellow).**

The housing and the selector both contain six teeth on the left and right sides. Therefore, the selector's translation in the positive or negative x-direction results in a 30° rotation around its x-axis as the selector slides over half the pitch distance of the teeth.

We designed the actuation unit using Solidworks (Dassault Systems Solidworks Corporation; Waltham, MA, USA). To facilitate the manual actuation of the selector, we added a translation ring to the actuation unit (Fig 6, Part 4 in red). The operator drives the translation ring with a reciprocating translating motion. Cylindrical pins on the translation ring interact with a circumferential slot in the selector, transmitting the translating motion in the x-direction while allowing the selector to rotate without the need for the operator's hand to rotate.

The needle segments run through the actuation unit at a larger diameter than at the needle tip. In order to guide the needle segments smoothly from the actuation unit to the needle tip, a double cone (blue) was designed at the distal side of the actuation unit. The double cone gently decreases the distance between the needle segments by guiding them smoothly through S-

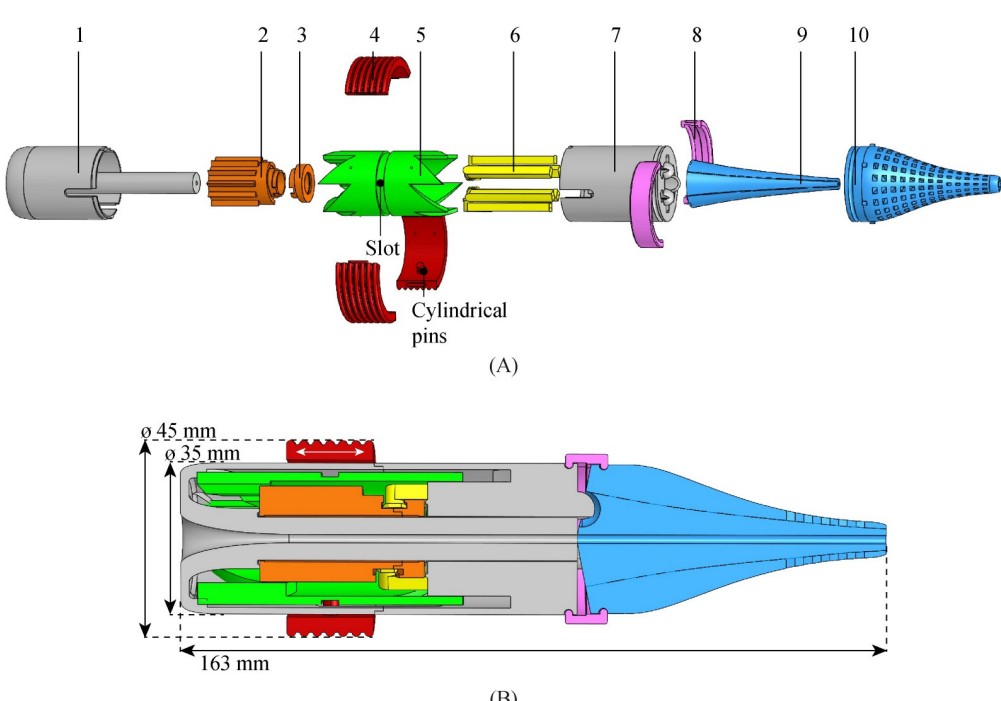

**Fig 6. Exploded view (A) and cross-section (B) of the actuation unit of the Ovipositor MRI-Needle, consisting of a housing bottom (1), cam bottom (2), cam top (3), translation ring (4), selector (5), needle segment holder (6), housing top (7), lock ring (8), inner double cone (9), and outer double cone (10).**

shaped channels from the actuation unit to the needle tip. These channels allow the needle segments to move back and forth freely while avoiding buckling.

## 2.3 Prototype manufacturing

**Material selection.** The American Society for Testing and Materials (ASTM) F2503 standard distinguishes three classifications for medical devices in the MR environment: MR-safe, MR-conditional, and MR-unsafe [32]. MR-safe devices are composed of electrically non-conductive, non-metallic, and non-magnetic materials; these devices are inherently safe to use in an MR environment [32]. Additionally, MR compatibility indicates the usability of the device in an MR environment, including potential image quality issues introduced by the device, according to ASTM F2119 [33].

**Needle.** For use inside an MRI system, the Ovipositor MRI-Needle must be at least MR conditional and MR compatible. Nitinol is metallic and, therefore, MR conditional at best. Nitinol is paramagnetic, meaning an external magnetic field weakly magnetises it while it loses its magnetism when the external magnetic field is removed [34]. Nitinol has a lower magnetic susceptibility than stainless steel; hence it produces fewer image artefacts than stainless steel [35, 36]. Therefore, medical devices made from Nitinol are frequently used in MRI-guided clinical procedures [37, 38]. The susceptibility difference between a Nitinol needle and the surrounding tissue may give rise to signal voids (due to strong $T2^*$ related signal decay) in the vicinity of the needle, which can be exploited as visualisation of the Nitinol needle [38, 39].

To comply with the diameter of conventional optical biopsy needles and optical treatment fibre positioning, the needle in this study consists of six 0.25-mm diameter rods, i.e., the needle segments. The needle segments are superelastic straight annealed Nitinol rods with a diameter

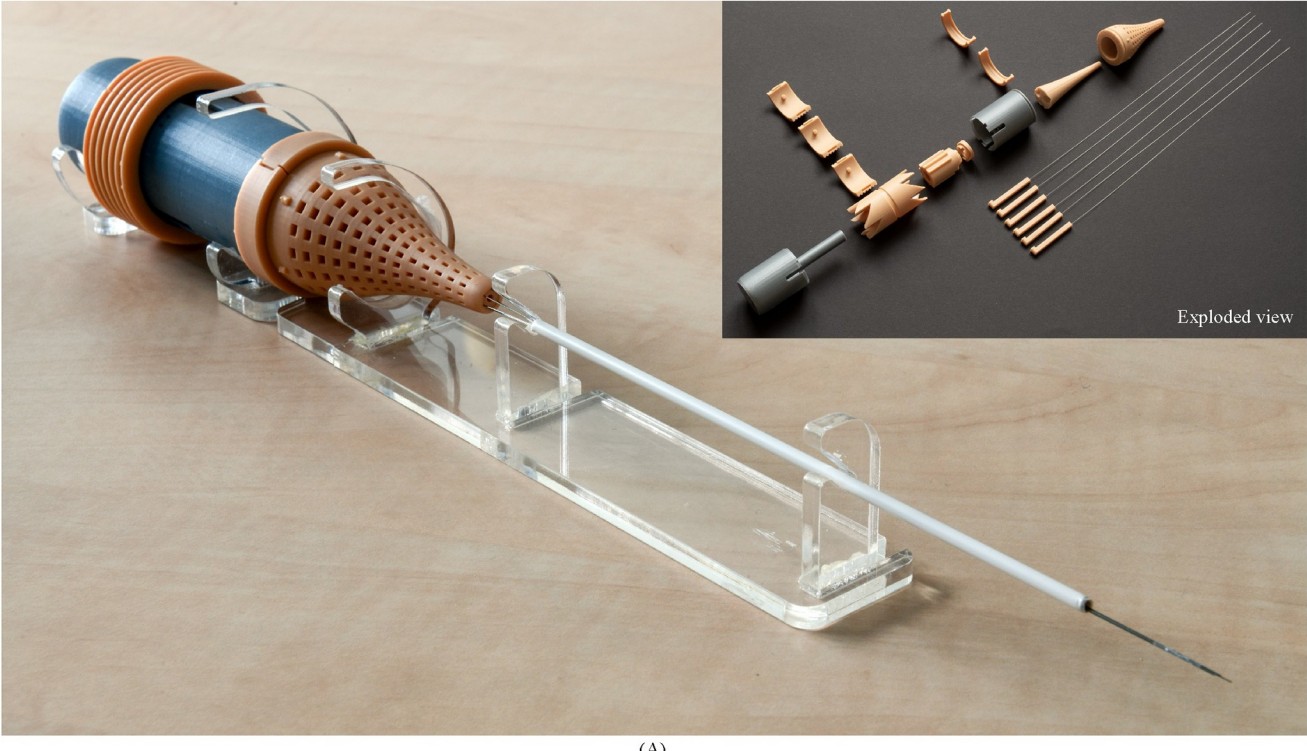

(A)

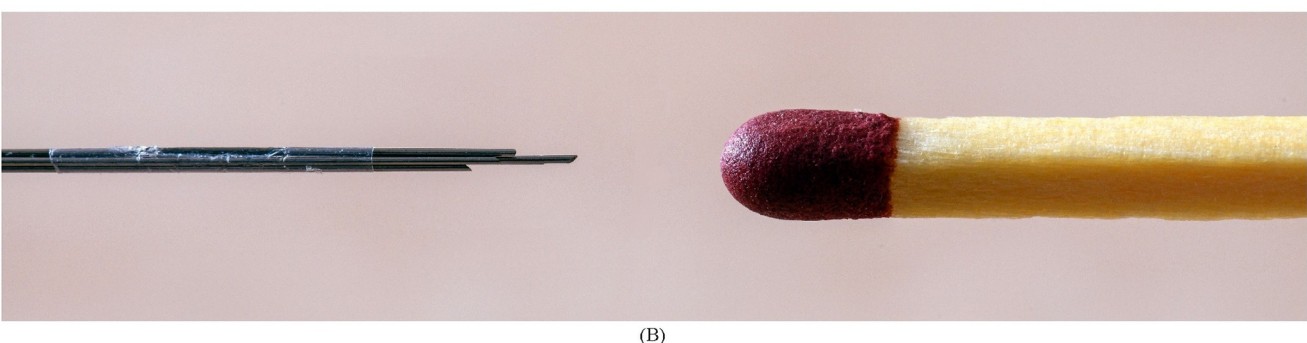

(B)

**Fig 7.** Ovipositor MRI-Needle (A) Close-up of the needle tip consisting of six sharpened Nitinol rods held together by a shrinking tube (Vention Medical) glued to one of the six rods. (B) Assembled prototype. The grey parts, i.e., the housing components, were produced using fused deposition technology (FDM) in polylactic acid (PLA) on an Ultimaker 3 printer. The orange parts, i.e., the actuation unit's internal components and the inner and outer double cones, were produced using stereolithography (SLA) technology in Dental Model resin V2 (Formlabs) on a Formlabs Form 3B printer. A transparent polymethyl methacrylate (PMMA) support structure supports the prototype, and a white polylactic acid (PLA) guide tube supports the needle in the support structure.

of 0.25 mm and a length of 276 mm, of which we placed 76 mm inside the actuation unit and 200 mm outside. The needle length is 200 mm to reach the prostate transperineally [30]. We glued the Nitinol rods (Pattex instant glue, Gold Gel) inside the needle segment holders (Fig 7). The cyanoacrylate-based glue is inherently biocompatible [40]. Fig 7A shows the tips of the needle segments, sharpened to an angle of 40° with wire electrical discharge machining. A 10-mm long shrinking tube (Vention Medical, expanded inner diameter 0.814 mm, wall thickness 0.013 mm) holds the six Nitinol rods together at the tip to limit the diverging of the needle segments while only minimally increasing the needle diameter. The shrinking tube is glued (Pattex instant glue, Gold Gel) to one of the needle segments to maintain its position at the

needle tip. The remaining needle segments can move freely back and forth through the shrinking tube while the needle segments are bundled at the tip. The resulting diameter of the needle, including the shrinking tube, is 0.84 mm.

**Actuation unit.** We used three-dimensional (3D) printing for the production of the actuation unit. The double cone of the actuation unit consists of two parts, an inner and an outer part, containing external and internal semi-circular grooves, respectively. We composed the double cone out of two parts with semi-circular grooves rather than one part with circular channels to prevent closing off those channels while using the stereolithography (SLA) 3D-printing process. Fig 6A shows that the double cone's outer part contains a hive structure. The hive structure leads to short grooves and short horizontal bridges across the grooves. A bridge is a material that links two raised points. Long bridges are likely to fail during the 3D-printing process or break during the post-processing of the 3D-printed component, thus closing off the needle grooves. The hive structure facilitates the 3D-printing process by creating short bridges that will not fail [41].

Fig 7B shows the 3D-printed components of the actuation unit as well as the assembled prototype. The components of the actuation unit were 3D printed on two different 3D printers, an SLA printer and a fused deposition modelling (FDM) printer. Two printers with different print settings with respect to the layer height were used to allow for a smooth gliding motion of the selector inside the housing. If the selector and housing were printed with the same layer height, they would fit together like puzzle pieces, leading to jamming of the selector inside the housing. The components of the actuation unit were 3D printed with Formlabs and Ultimaker 3D printers, using Dental Model V2 resin (Formlabs) and polylactic acid (PLA), respectively, both materials being MR safe. We printed the SLA parts using a Formlabs Form 3B printer with a layer height of 0.050 mm, and the FDM parts using an Ultimaker 3 printer with a layer height of 0.1 mm. During assembly, we glued the housing bottom and housing top together (Pattex instant glue, Gold Gel). The height of the cam track dictates a 4-mm stroke in the positive x-direction for the needle segment holders over a 60˚ rotation of the cam. During the following 300˚ rotation, the cam track dictates in steps a total of 4-mm stroke in the negative x-direction. The actuation unit's length is 163 mm (Fig 6B), the outer diameter is 35 mm, and the outer diameter of the translation ring is 45 mm. The hollow core has a diameter of 2 mm.

## 3. Evaluation

### 3.1 Goal of the experiment

In a proof-of-principle experiment, the functioning of the developed Ovipositor MRI-Needle was evaluated in *ex vivo* human prostate tissue inside a preclinical 7-T MRI system (MR Solutions, Guildford, United Kingdom) at the Amsterdam University Medical Center (AUMC, department of Biomedical Engineering and Physics). The discarded human prostate sample was collected anonymously at the Amsterdam University Medical Center (www.amsterdamu mc.org). The human prostate sample came from a deceased patient who had approved to donate his body to science, and his prostate was collected after autopsy. Therefore, this experiment was not subject to the Medical Research Involving Human Subjects Act (WMO), and it did not have to undergo review by an accredited Medical Research Ethics Committee or the Central Committee on Research Involving Humans.

We evaluated the performance of the Ovipositor MRI-Needle in terms of the slip of the needle with respect to the prostate tissue. More specifically, we calculated the slip ratio over an

entire measurement as in Eq 2:

$$s_{\text{ratio}} = 1 - \left(\frac{d_{\text{m}}}{d_{\text{e}}}\right) \tag{2}$$

where $d_{\text{m}}$ and $d_{\text{e}}$ are the measured and expected travelled distance, respectively. The expected travelled distance is 4 mm in one cycle due to the 4-mm stroke dictated by the cam track. In one cycle, all six needle segments are advanced in one step and retracted in five steps, meaning that one cycle equals a full rotation of the cam. For one cycle, we had to translate the translation ring for 12 repetitions. During one measurement, the needle was actuated for ten cycles, i.e., 120 translations of the translation ring, which means that the total expected travelled distance during one cycle was equal to 40 mm. The measured travelled distance is the difference in the position of the needle tip we measured in the MR images before and after the needle actuation for ten cycles.

## 3.2 Experimental setup

Fig 8 shows the experimental setup, consisting of the Ovipositor MRI-Needle, an *ex vivo* human prostate tissue sample embedded in agar in a tissue box, and a preclinical 7-T MRI system (MR Solutions, Guildford, United Kingdom). Instead of moving a needle towards the tissue, we decided to move the tissue in the tissue box towards the needle. Specifically, we kept the actuation unit stationary, fixed to the MRI system, to use the manual actuation force solely for the translation of the needle segments with zero external push force. The principle of needle insertion with zero external push force holds if the self-propelled needle pulls the tissue towards itself by pulling itself deeper into the tissue, thereby pulling the tissue box towards the needle. The MRI system contained a half-round tube that could be slid into and out of the housing of the MRI system. On top of the half-round tube, a radio frequency (RF) coil was positioned. Inside the RF coil, the tissue sample can be positioned to allow for visualisation using MRI acquisitions. During the performance evaluation of the Ovipositor MRI-Needle, we were interested in the position of the needle tip; therefore, the tissue box was placed inside the RF coil. The RF coil has an inner radius of 65 mm. In order to test the Ovipositor MRI-Needle inside the MRI system, we needed a movable support structure for the tissue box to allow low-friction horizontal translation of the tissue box inside the RF coil while constraining the rolling motion in the lateral direction. The low-friction structure consisted of box rails attached to the tissue box, an RF base plate attached to the RF coil, and wheels between the box rails and RF base plate; for more details, see S1 Appendix.

We prepared the biological sample by placing a piece of *ex vivo* human prostate tissue (width 25 mm, length 50 mm, height 10 mm) in a preparation box with liquid agar (2.5%wt). Storing the box in the refrigerator overnight, fixated the tissue in the agar. We cut the sample to the correct dimensions (width 50 mm, length 90 mm, height 10 mm, weight 52 g) to fit inside the tissue box used in the experiment, with the prostate tissue at the tissue's box distal end aligned with a central hole in the wall of the box for insertion of the needle. The remaining part of the tissue box filled with agar allowed an initial insertion depth of 40 mm of the needle into the agar before entering the prostate tissue. To enable multiple slip ratio measurements in a single prostate tissue sample, the box contained multiple holes for insertion of the needle, allowing testing in different parts of the prostate sample with a new needle trajectory for each measurement. The insertion holes were placed at a distance of 2.5 mm from each other to avoid overlapping needle trajectories, resulting in a total of seventeen holes numbered from one to seventeen from left to right (Fig 8C). We positioned the needle in front of the tissue so it could move through the tissue. Unfortunately, because of tissue inhomogeneities or the

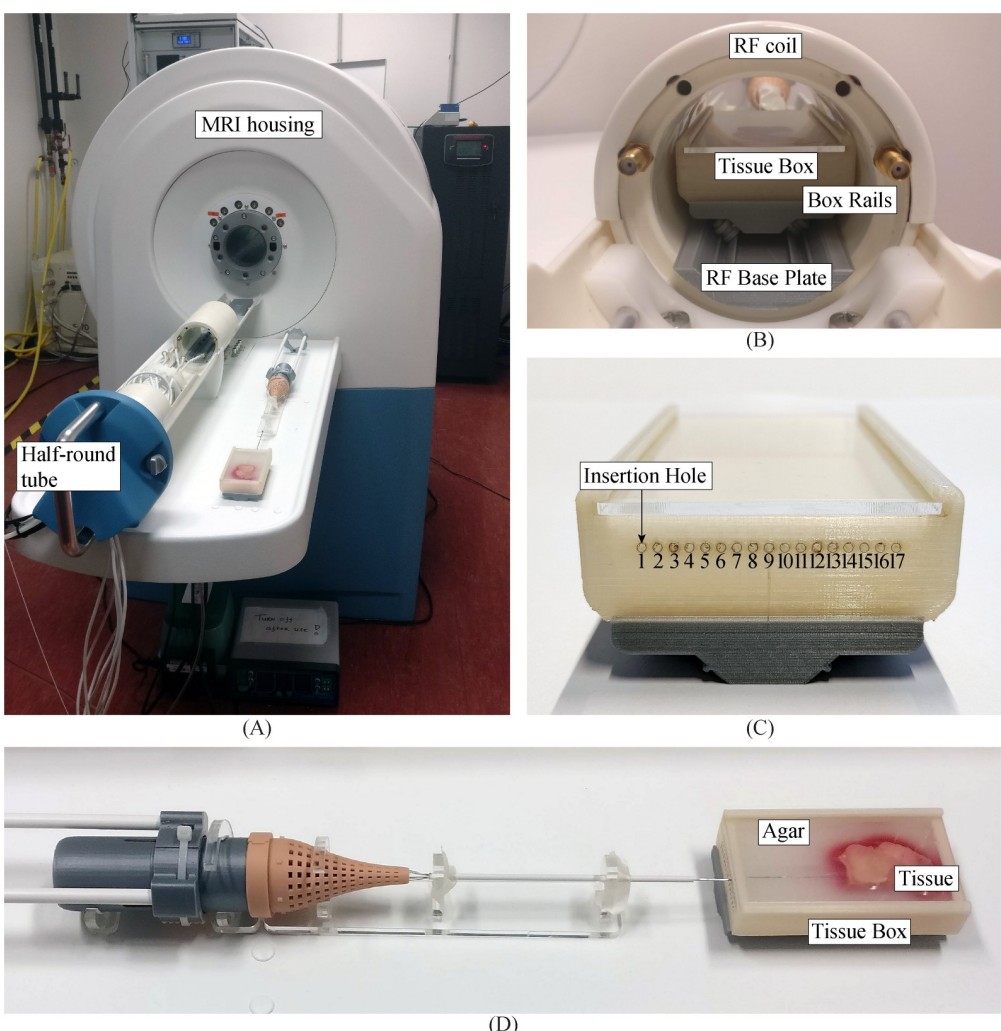

**Fig 8. Experimental setup of the *ex vivo* prostate tissue experiment.** (A) The instrument was placed in a half-round tube with a support structure in between. The half-round tube was slid into the MRI bore. (B) Close-up of the radiofrequency (RF) coil with the tissue box on the RF base plate, guided on rails. (C) Close-up of the proximal side of the tissue box containing seventeen insertion holes. (D) Close-up of the *ex vivo* prostate tissue embedded in solidified 2.5%wt agar, with the needle inserted through the agar in front of the tissue.

absence of tissue behind the holes, experiments could only be carried out in a few specific holes (no. 6, 7, and 9). The tissue behind these specific holes appears homogenous on the MR images. Hence, we assumed the behaviour of the needle to be the same for the experiments using these holes.

### 3.3 Experimental procedure

In our experiment, we used 3D gradient-echo acquisitions to capture the needle position with respect to the prostate tissue and the tissue box (see S2 Appendix for imaging parameters). We conducted three measurements using three different insertion holes in a randomised order. During a single measurement, the Ovipositor MRI-Needle was actuated for ten actuation cycles, i.e., 120 translations of the translation ring. For every measurement, a 3D gradient echo acquisition captured the static needle position at the start and after the ten actuation cycles. S3

Appendix contains a detailed explanation of the steps in the experimental protocol. After each measurement, we cleaned the needle with water and alcohol. All measurements were conducted in one day.

## 3.4 Experimental results

3D gradient-echo acquisitions visualised the start and end position of the needle tip. Fig 9 shows the MR images of the needle tip positions (Original MR images can be found in S4 Appendix). Table 1 shows the insertion depth and travelled distance measurements. The initial insertion depth was that of the needle tip when it was positioned in front of the prostate tissue. The travelled distance was the distance the needle travelled inside the prostate tissue. The

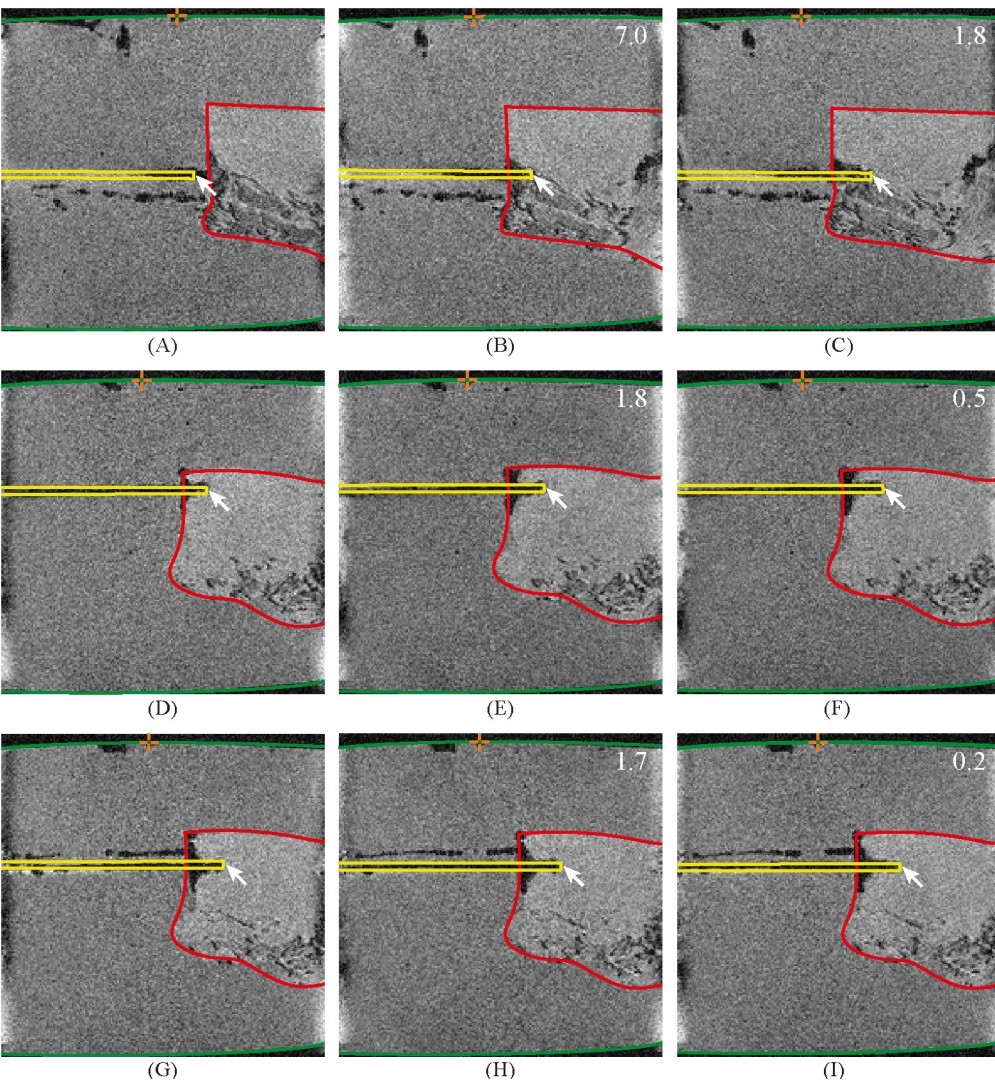

**Fig 9. MR images of the needle inside the agar and *ex vivo* prostate tissue.** Each row represents one measurement. The first column shows the initial frame where the tip is positioned inside the agar in front of the prostate tissue. The second column shows the frame after actuation over five cycles. The third column shows the frame after the second actuation over five cycles. The yellow, red, and green contours show the needle, the prostate tissue, and the tissue box sides, respectively. The arrow marks the needle tip. The orange crosshair shows the reference point on the side of the box that indicates a 40-mm insertion depth. The number in the upper right corner in black shows the measured travelled distance of the tissue box, $d_m$, in mm, with respect to the previous MR image.

**Table 1. Results of the *ex vivo* evaluation.**

| Measurement | Insertion hole | Initial insertion depth [mm] | Measured travelled distance, $d_m$ [mm] | Cycles | Expected travelled distance, $d_e$ [mm] | Slip ratio, $s_{ratio}$ |
|---|---|---|---|---|---|---|
| 1 | 9 | 44.1 | 8.8 | 10 | 40 | 0.78 |
| 2 | 6 | 50.9 | 2.3 | 10 | 40 | 0.94 |
| 3 | 7 | 51.9 | 1.9 | 10 | 40 | 0.95 |

For each measurement, the following information is reported: the insertion hole used on the tissue box, measured travelled distance [mm] of the box, number of cycles needed to travel that distance, expected travelled distance [mm] that the box would have travelled if no slip occurred, and the slip ratio.

needle was able to propel itself through the prostate tissue. However, the needle did experience slip. We measured a slip ratio in the range of 0.78–0.95. This slip indicates that the needle segments unintentionally slide with respect to the tissue, resulting in a shorter measured travelled distance than the expected travelled distance.

## 4. Discussion

### 4.1 Needle performance

This study reported on the design and experimental validation of a self-propelled needle with an MRI-ready actuation system. The evaluation of the needle in *ex vivo* prostate tissue showed the needle was able to advance through the tissue. However, the needle did experience a high slip ratio. We measured a slip ratio in the range of 0.78–0.95. The slip ratio of our needle in *ex vivo* prostate tissue is comparable to that of the self-propelled needle developed by Scali [18], who reported a slip ratio in the range of 0.87–0.90 for the continuously moving needle in *ex vivo* porcine liver tissue. The high slip ratio in our measurements indicates that the cutting and friction forces acting on the advancing needle segment and the friction forces on the wheels of the support structure altogether were near the friction forces on the retracting needle segments (Eq 1). Furthermore, despite the shrinking tube, the needle segments diverged a little at their tips like an opening umbrella during the experiment, hindering the needle segments' advancing motions, thereby increasing the needle's slip ratio. The degree to which the segments diverged at the needle tip could not be quantified because of the low resolution of the MRI system used.

### 4.2 Limitations

The needle segments were designed and sharpened so that the needle segments point toward the middle of the needle. However, the needle segments did not always point toward the middle of the assembled prototype. This could be explained by the way the needle segments are bundled. Along the length of the 200-mm needle, the needle segments were only connected at the tip by a 10-mm long shrinking tube. Hence, the needle segments could change position during the experiment, causing the needle segments to rotate and entangle. Because the needle segments did not point toward the middle, their bevel-shaped tips might have caused them to diverge and allow tissue accumulation between them. In future designs, we aim to point all bevel-shaped tips toward the middle of the needle tip by restricting their rotation to prevent the needle segments from diverging.

At the needle tip, the six needle segments are surrounded by the shrinking tube, which might hinder the needle's self-propelling motion through the tissue. However, as the needle is advanced further into the tissue, the surface area of the needle segments in direct contact with the tissue increases, whereas the surface area of the shrinking tube in contact with the tissue

remains unchanged. Consequently, the influence of the shrinking tube on the self-propelling motion declines as the needle advances further into the tissue. In a future version of the Ovipositor MRI-Needle, we will investigate methods to replace the shrinking tube with another connection mechanism to improve the needle's self-propelling motion.

### 4.3 Future work

In this study, we placed the tissue in a box that moved towards the needle. For application in transperineal laser ablation, the needle will have to self-propel through the perineal skin and into the prostate while the patient stays still inside the MRI bore. The actuation unit can be placed on a robotic arm suited to move the needle towards the cancerous tissue. The current Ovipositor MRI-Needle uses a discrete manual translating motion of the translation ring as its input. In future work, the translation ring could be replaced with a motorised actuation unit that is safe for use in an MRI environment. Electric motors are not an option due to the interference of these motors with the magnetic field. Alternative actuation methods are piezo motors, ultrasonic motors, Bowden cables, pneumatics, hydraulics, magnetic spheres, and shape memory alloy actuators. In a hospital setup, pneumatics are advantageous as pressurised air is commonly available in an MRI room. An important drawback of pneumatics is that air is compressible, so the only well-defined pneumatic actuator positions are the beginning and end positions [42]. This makes pneumatic actuators more suited for a discrete stepwise motion instead of continuous motion. Our selector mechanism is currently actuated by a stepwise manual translation, which can relatively easily be replaced by a stepwise pneumatic actuation mechanism.

In our experiment, the performance evaluation of the Ovipositor MRI-Needle was limited to an evaluation in a single *ex vivo* frozen-thawed prostate sample embedded in agar. Larger sample sizes are needed for future evaluations, considering that the mechanical properties of the prostate tissue of different men are not the same but comprise ranges of values.

Another limitation was that the sample was frozen quickly at -80 ºC in liquid nitrogen and thawed prior to the experiment. While rapid freezing reduces ice crystal formation in the tissue and minimizes morphological changes [43], Venkatasubramanian *et al.* [44] demonstrated that freezing tissue in liquid nitrogen could affect tissue stiffness compared to fresh tissue, with the exact effects of this freeze-thaw cycle on the mechanical properties of the tissue being still unknown. However, this effect might not influence the needle's self-propelling motion as in our experiment, the Ovipositor MRI-Needle could self-propel through tissue that had been frozen. In comparison, Scali [18] evaluated the performance of a wasp-inspired self-propelled needle in *ex vivo* four hours post mortem porcine tissue that did not undergo a freeze-thaw cycle and showed that the needle could self-propel through relatively fresh *ex vivo* tissue with a comparable slip ratio as the Ovipositor MRI-Needle in *ex vivo* prostate tissue.

Another limitation related to our experiment conditions, the agar (2.5%wt) in which the prostate tissue was embedded was stiffer than the prostate tissue itself, which could have affected the self-propelling motion of our Ovipositor MRI-Needle. Scali *et al.* [27] showed that the slip ratio of a self-propelled needle was higher in tissue stiffer compared to less stiff phantoms. This indicates that the stiff agar in our experiment could have increased the slip ratio of the Ovipositor MRI-Needle compared to when the needle would advance through prostate tissue.

In future studies, *in vivo* experiments are needed to test the performance of the needle in a more realistic clinical scenario. When moving toward an *in vivo* study, we foresee some challenges, such as the imaging system, the presence of blood flow through the prostate gland, and the presence of multi-layered tissue. For *in vivo* tests inside porcine animal models, we need an MRI system with a bore and RF coil that fits the animal model, as the currently used preclinical MRI system has an RF coil diameter of only 65 mm. Alternatively, other imaging options like

ultrasound could be used, which will have their own advantages and disadvantages, such as low contrast for soft tissues [45]. The presence of blood flow could decrease the needle-tissue friction required for the self-propelling motion. However, the parasitic wasp is able to advance through more liquid substrates such as fruits (e.g., figs) [46, 47] thanks to harpoon-like serrations on the valves, which increase friction [47]. Similarly, a microtextured directional surface topography could be added to the needle surface, as shown by Parittotokkaporn *et al.* [48]. Another challenge is that in an *in vivo* model, there is more and multi-layered tissue between the insertion point (i.e., the perineum) and the target position inside the prostate gland. However, other self-propelled needles have been shown to be able to advance in multi-layered tissue-mimicking phantoms consisting of gelatine with different stiffnesses [27]. Moreover, when the needle is inserted through multiple layers of tissue and thus deeper into the tissue, the self-propelling motion is expected to work better, as the role of the cutting force of the single advanced segment becomes less pertinent compared to the friction forces of the retracted segments.

In this study, we kept the actuation unit stationary while the box was placed on a low-friction support structure that moved towards the needle. In clinical practice, the needle will have to self-propel inside the patient while the tissue remains in place. The actuation unit could be placed on a dedicated robotic arm to manipulate the needle towards the patient, following the pace of the self-propelling motion of the needle. Moreover, for the future production of the needle segments, industrial needle manufacturing processes could be used to produce a needle with a sharper tip (e.g., a lancet point) to facilitate the propulsion through the tissue.

Currently, clinicians typically need multiple attempts to reach the target location, leading to an increased risk of tissue damage [7]. Moreover, a narrow pubic arch or a large prostate can obstruct the needle trajectory, making it difficult to reach certain prostate locations [49]. Steerable needles can help the clinician compensate for positioning errors and follow a curved path to reach all positions inside the prostate while avoiding anatomical obstacles. In a future version of the Ovipositor MRI-Needle, a steering mechanism can be incorporated. Steering can be achieved by creating an offset between the needle segment tips to approximate a bevel-shaped tip. The surrounding tissue exerts forces on the bevel-shaped tip in an asymmetric fashion, resulting in the bending of the needle in the direction of the bevel [50]. Research by Scali *et al.* [26] on wasp-inspired steerable needles showed that approximated bevel-shaped tips could be used to steer the needle successfully. Research into steering will be incorporated in future prototypes of our needle.

Considering the needle's primary goal, its use in MRI-guided transperineal optical biopsy and focal laser ablation, the MR safety and compatibility of the components of the Ovipositor MRI-Needle should in the future be addressed using the ASTM test methods [32]. Radiofrequency heating caused by the Nitinol needle should be evaluated experimentally, as the Nitinol needle is a long and electrically conductive structure that couples with the electric field of the RF coil in an MRI system [51]. The coupling induces high voltages at the end of the needle, which might cause heating of the surrounding tissue that poses a potential safety hazard to the patient [51, 52]. Alternatively, the Nitinol needle segments could be replaced by needle segments made of electrically non-conductive, non-metallic, and non-magnetic materials such as polymer needle segments or glass fibres. An MRI-ready, self-propelled, steerable needle can serve as a platform technology for the precise positioning of a functional element in a target region in the body.

## 5. Conclusion

This work presents the design and experimental validation of a self-propelled needle with an MRI-ready actuation system. We have shown that a discrete manual translating motion can

actuate the reciprocating motion of the six parallel needle segments using a selector. A continuous hollow core through the actuation unit allows for needle functionalisation with an optical fibre for optical biopsy and focal laser ablation. The prototype's components, excluding the needle, are easily manufactured solely by 3D printing using MR-safe materials. The needle consists of six sharpened Nitinol rods. It was possible to determine the needle tip's position in the MR image, as the Nitinol needle did not cause severe image artefacts. The evaluation of the prototype in *ex vivo* human prostate tissue in an MRI system showed that the needle was able to self-propel through the tissue. However, it experiences a high slip ratio. The Ovipositor MRI-Needle is a step forward in developing a self-propelled needle for MRI-guided transperineal focal laser ablation to treat prostate cancer.

## Supporting information

**S1 Appendix. Support structure design and manufacturing.**
(DOCX)

**S2 Appendix. Magnetic resonance parameters.**
(DOCX)

**S3 Appendix. Experimental protocol.**
(DOCX)

**S4 Appendix. Study data and original MR images.**
(DOCX)

**S1 Video. Video of the needle tip moving the needle segments subsequently.**
(MP4)

## Acknowledgments

We would like to thank David Jager from DEMO (Dienst Elektronische en Mechanische Ontwikkeling) at the TU Delft for sharpening the needle segments and Daniel Martijn de Bruin, Luigi van Riel, Gustav Strijkers, and Theo de Reijke from the AUMC (Amsterdam University Medical Centers) for the discussions about the application of the project in focal laser ablation for prostate cancer treatment and their help in designing the setup and performing the experiment.

## Author Contributions

**Conceptualization:** Jette Bloemberg, Fabian Trauzettel, Dimitra Dodou, Paul Breedveld.

**Data curation:** Jette Bloemberg, Fabian Trauzettel, Bram Coolen.

**Investigation:** Jette Bloemberg, Fabian Trauzettel.

**Methodology:** Jette Bloemberg, Fabian Trauzettel, Bram Coolen, Dimitra Dodou, Paul Breedveld.

**Validation:** Jette Bloemberg.

**Writing – original draft:** Jette Bloemberg.

**Writing – review & editing:** Jette Bloemberg, Fabian Trauzettel, Bram Coolen, Dimitra Dodou, Paul Breedveld.

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
