## [Decision Letter · Decision Letter 0]

6 Jul 2022

PONE-D-22-12859Design and evaluation of an MRI-ready, self-propelled needle for prostate interventionsPLOS ONE

Dear Dr. Bloemberg,

Thank you for submitting your manuscript to PLOS ONE. After careful consideration, we feel that it has merit but does not fully meet PLOS ONE’s publication criteria as it currently stands. Therefore, we invite you to submit a revised version of the manuscript that addresses the points raised during the review process.

Specifically, the authors should add more details related to the actuation method chosen and foreseen challenges in moving towards in-vivo characterizations.

We look forward to receiving your revised manuscript.

Kind regards,

Tommaso Ranzani, PhD

Academic Editor

PLOS ONE

Journal Requirements:

Reviewers' comments:

Reviewer's Responses to Questions

**Comments to the Author**

1. Is the manuscript technically sound, and do the data support the conclusions?

Reviewer #1: Yes

Reviewer #2: Yes

2. Has the statistical analysis been performed appropriately and rigorously? 

Reviewer #1: No

Reviewer #2: Yes

3. Have the authors made all data underlying the findings in their manuscript fully available?

Reviewer #1: No

Reviewer #2: Yes

4. Is the manuscript presented in an intelligible fashion and written in standard English?

Reviewer #1: Yes

Reviewer #2: Yes

5. Review Comments to the Author

Reviewer #1: The authors present an ingenious bio-inspired needle with six segments. It can be manually actuated by a cam mechanism that advances one of of the segments at a time.

The design is interesting and definitely worth pursuing.

The presentation is clear and the paper is well written.

The major limitation of the design is the lack of experimentation in vivo. The problem is that needle channels in living tissue are almost instantaneously lubricated with blood and further tissue damage is also accompanied by inflammation. This may pose difficulties for the approach, as the needle track risks becoming very slippery. The ex vivo prostate study is helpful but it does not present the full potential issue. To check this, the authors do not need to carry out the experiment in MRI; they could instead use ultrasound and an animal study at the local hospital. Porcine models are used all the time for teaching, so it will not be a huge effort to quantify performance of their device in living animal tissue, e.g. a porcine liver.

I would like to see some discussion of this observation in the paper and plans to address the potential problem of friction (or lack thereof).

Reviewer #2: In this work the authors present a design for a self-inserting needle with potential applications in prostate biopsy/ablation. The study experimentally evaluates the ability of the needle to insert itself, quantitatively measuring the slip ratio during deployment. The study was conducted in cadaveric human prostate tissue embedded in an agar gel phantom.

The paper is very well-written and clearly presents the design, evaluation study, and results. The reviewer has a few comments and suggestions for improvement. In no particular order:

There have also been recent works in other types of needles that are pulled from their tip rather than pushed from their base. For instance, [1] leverages magnetic force to pull the needle, and recently [2] leverages a screw-tip mechanism to pull the needle and magnetic torque to steer the needle. The authors may consider mentioning designs such as these which, while significantly different, may exhibit similar benefits as the proposed design:

[1] M. Ilami, R. J. Ahmed, A. Petras, B. Beigzadeh, and H. Marvi, “Magnetic Needle Steering in Soft Phantom Tissue”, Scientific Reports, 10(1), pp. 1-11, 2020.

[2] T. J. Schwehr, A. J. Sperry, J. D. Rolston, M. D. Alexander, J. J. Abbott, and A. Kuntz, "Toward Targeted Therapy in the Brain by Leveraging Screw-Tip Soft Magnetically Steerable Needles", Hamlyn Symposium on Medical Robotics, pp. 81-82, 2022.

In Equation 2, d_m and d_e are used, however when describing the equation, the authors define d_t (line 280). This discrepancy should be resolved.

Do the authors have any intuition regarding the differences between the agar used in the study to surround the ex vivo prostate tissue and the tissue in the human body that would represent intermediate tissue the needle must move through en route to the prostate? It may be worth briefly discussing this point and its implications for the study/results. E.g., is it possible the agar may have exaggerated the slip in this case compared with human tissue?

How does the shrinking tube that keeps the individual needle segments together near the tip impact the working principle of the self-propelled insertion? Does the shrinking tube hinder the insertion of the needle? If it affects this, to what expected degree? It may be worth discussing this point as well.

Can the authors quantify the degree to which the needles diverged at their tips during the experiments? Doing so would add significant context to the observation.

In lines 395-396 in the discussion section, the authors state “This makes pneumatic actuators more suited for a discrete, stepwise motion instead of a continuous motion by using our selector mechanism, for example.” This is unclear. Are the authors stating that their mechanism is an example of a continuous motion that does not suit itself to pneumatic actuators or are they instead stating the proposed mechanism is an example of a discrete motion? The text is ambiguous. However, it is my impression that the proposed needle would be an example that can well be actuated by stepwise pneumatic mechanisms.

The authors mention the concern of heating of the nitinol in the MRI. It seems to me, however, that nitinol is not necessary as the material for the needle bundle. If the design could instead, theoretically, incorporate non-magnetic materials in the needle bundle, it may be worth mentioning this in the future work section when bringing up the heating concern.

It may be an artifact of the review process, or fixed later in the editorial process, but it is worth noting that many of the figures exhibit quite low resolution.

A few typos noted:

Line 109 prostat…e

Line 214 or visualization

The paper is clearly presented and quite well written. If the above comments were addressed the paper would be further strengthened.

6. PLOS authors have the option to publish the peer review history of their article (what does this mean?). If published, this will include your full peer review and any attached files.

Reviewer #1: No

Reviewer #2: No

---

## [Author Response · Author response to Decision Letter 0]

26 Jul 2022

Dear Tommaso Ranzani, 

Thank you for contacting the reviewers. We have revised the manuscript accordingly. This letter was also uploaded as an attachment.

Regarding the points as described in your email, we confirm that (1) we adapted the manuscript to meet PLOS ONE’s style requirements, (2) we provided the correct grant numbers for our study in the ‘Funding Information’ section in the submission system and we added an updated financial disclosure statement in our cover letter, (3) we uploaded our study’s minimal underlying data as Supporting Information file 4 (i.e., S4 Appendix. Study data and original MR images), (4) our ethics statement only appears in Section 3, and (5) we reviewed our reference list to make sure that it is complete and correct. 

Regarding the details related to the actuation method chosen and foreseen challenges in moving towards in vivo testing, we added a clarification in Section 4.3. In the first paragraph of Section 4.3, we explain that the manual actuation of the selector mechanism of the Ovipositor MRI-needle could be replaced by a stepwise pneumatic actuation mechanism (see also R7). We also added two paragraphs in Section 4.3 about the foreseen challenges for executing in vivo testing and the required improvements to the Ovipositor MRI-Needle before such testing becomes possible (see also R1). 

Please find our detailed responses to the reviewers below. In the revised manuscript, the changes are highlighted.

Review Comments to the Author

Reviewer #1: The authors present an ingenious bio-inspired needle with six segments. It can be manually actuated by a cam mechanism that advances one of of the segments at a time.

The design is interesting and definitely worth pursuing.

The presentation is clear and the paper is well written.

The major limitation of the design is the lack of experimentation in vivo. The problem is that needle channels in living tissue are almost instantaneously lubricated with blood and further tissue damage is also accompanied by inflammation. This may pose difficulties for the approach, as the needle track risks becoming very slippery. The ex vivo prostate study is helpful but it does not present the full potential issue. To check this, the authors do not need to carry out the experiment in MRI; they could instead use ultrasound and an animal study at the local hospital. Porcine models are used all the time for teaching, so it will not be a huge effort to quantify performance of their device in living animal tissue, e.g. a porcine liver.

I would like to see some discussion of this observation in the paper and plans to address the potential problem of friction (or lack thereof).

R1. Thank you for your compliments. In Subsection 4.3, we now discuss the foreseen challenges in moving towards in vivo testing and the required adaptations to the prototype and experimental setup before such testing becomes possible. The following challenges regarding the tissue are mentioned: (1) the blood flow through the prostate gland and (2) more and multi-layered tissue between the insertion point (i.e., the perineum) and the target location inside the prostate. Regarding the experimental setup we discuss the MRI scanner that should fit the in vivo animal model. We discuss the following additional developments required for the Ovipositor MRI-Needle: (1) a robotic arm to move the needle towards the tissue instead of the tissue towards the needle and (2) further sharpening of the needle segments. Furthermore, we also discuss the potential problem of friction, how the parasitic wasp solves this problem and how this solution could be implemented in the needle. 

Reviewer #2: In this work the authors present a design for a self-inserting needle with potential applications in prostate biopsy/ablation. The study experimentally evaluates the ability of the needle to insert itself, quantitatively measuring the slip ratio during deployment. The study was conducted in cadaveric human prostate tissue embedded in an agar gel phantom.

The paper is very well-written and clearly presents the design, evaluation study, and results. The reviewer has a few comments and suggestions for improvement. In no particular order:

There have also been recent works in other types of needles that are pulled from their tip rather than pushed from their base. For instance, [1] leverages magnetic force to pull the needle, and recently [2] leverages a screw-tip mechanism to pull the needle and magnetic torque to steer the needle. The authors may consider mentioning designs such as these which, while significantly different, may exhibit similar benefits as the proposed design:

[1] M. Ilami, R. J. Ahmed, A. Petras, B. Beigzadeh, and H. Marvi, “Magnetic Needle Steering in Soft Phantom Tissue”, Scientific Reports, 10(1), pp. 1-11, 2020.

[2] T. J. Schwehr, A. J. Sperry, J. D. Rolston, M. D. Alexander, J. J. Abbott, and A. Kuntz, "Toward Targeted Therapy in the Brain by Leveraging Screw-Tip Soft Magnetically Steerable Needles", Hamlyn Symposium on Medical Robotics, pp. 81-82, 2022.

R2. Thank you for your compliments and for pointing out these references. In Section 1.2 (Lines 59-65), we have now added the designs and their working mechanisms to steer and overcome buckling as described by Ilami et al. (2020) and Schwehr et al. (2022). 

In Equation 2, d_m and d_e are used, however when describing the equation, the authors define d_t (line 280). This discrepancy should be resolved.

R3. Thank you for pointing out this discrepancy. In Equation 2 and in the text, we have now changed d_t to d_e. 

Do the authors have any intuition regarding the differences between the agar used in the study to surround the ex vivo prostate tissue and the tissue in the human body that would represent intermediate tissue the needle must move through en route to the prostate? It may be worth briefly discussing this point and its implications for the study/results. E.g., is it possible the agar may have exaggerated the slip in this case compared with human tissue?

R4. Thank you for this suggestion. In the second paragraph of Subsection 4.3, we have now added an explanation of how the agar could have increased the slip ratio because of the high stiffness of the agar (2.5wt%) used in our experiment. Moreover, we added a third paragraph, where we discuss the foreseen challenges in moving towards in vivo testing (see also R1). In this new paragraph, we discuss the effects of more and multi-layered tissue between the insertion point of the needle in in vivo conditions (i.e., the perineum) and the target point inside the prostate. Other non-MRI-ready self-propelled needles have been shown to be able to self-propel in multi-layered tissue-mimicking phantoms [3]. Moreover, as the needle is inserted further into the tissue, the role of the cutting force on the single advanced segment becomes less pertinent, which in theory would aid the self-propelling motion of the needle. 

[3] Scali M, Breedveld P, Dodou D. Experimental evaluation of a self-propelling bio-inspired needle in single-and multi-layered phantoms. Scientific reports. 2019;9(1):1-13. doi: 10.1038/s41598-019-56403-0

How does the shrinking tube that keeps the individual needle segments together near the tip impact the working principle of the self-propelled insertion? Does the shrinking tube hinder the insertion of the needle? If it affects this, to what 

expected degree? It may be worth discussing this point as well.

R5. Thank you for this suggestion. We agree that an explanation of how the shrinking tube hinders the insertion mechanism of the needle is highly relevant. In Subsection 4.2, we added an explanation about how the needle’s self-propelling motion might be hindered by the increased friction force of the needle segment that is glued to the shrinking tube advances and how this effect declines as the needle further self-propels into the tissue. 

Can the authors quantify the degree to which the needles diverged at their tips during the experiments? Doing so would add significant context to the observation.

R6. Thank you for this suggestion. Unfortunately, the resolution of the MRI system used was not high enough to quantify the degree to which the needle tips diverged. We added this explanation in Subsection 4.1. Furthermore, in Subsection 4.2, we added a future design aim where we describe that we want to point all bevel-shaped tips toward the centre of the needle tip by restricting the rotation of the needle segment tips to prevent diverging of the needle segments. 

In lines 395-396 in the discussion section, the authors state “This makes pneumatic actuators more suited for a discrete, stepwise motion instead of a continuous motion by using our selector mechanism, for example.” This is unclear. Are the authors stating that their mechanism is an example of a continuous motion that does not suit itself to pneumatic actuators or are they instead stating the proposed mechanism is an example of a discrete motion? The text is ambiguous. However, it is my impression that the proposed needle would be an example that can well be actuated by stepwise pneumatic mechanisms.

R7. Thank you for pointing out this ambiguity. In Subsection 4.3, we now describe that a stepwise pneumatic actuation mechanism can replace the current stepwise manual translation. 

The authors mention the concern of heating of the nitinol in the MRI. It seems to me, however, that nitinol is not necessary as the material for the needle bundle. If the design could instead, theoretically, incorporate non-magnetic materials in the needle bundle, it may be worth mentioning this in the future work section when bringing up the heating concern.

R8. We completely agree that the needle segments in the design could be replaced by needle segments of non-magnetic materials. In Subsection 4.3, we added the design consideration of changing the needle segment materials. 

It may be an artifact of the review process, or fixed later in the editorial process, but it is worth noting that many of the figures exhibit quite low resolution. 

R9. We checked the resolution of the figures during the submission process of the revision. 

A few typos noted:

Line 109 prostat…e

Line 214 or visualization

R10. We removed these typos.

The paper is clearly presented and quite well written. If the above comments were addressed the paper would be further strengthened.

Thank you your useful suggestions. 

We look forward to your reaction.

With kind regards,

Jette Bloemberg and co-authors

---

## [Decision Letter · Decision Letter 1]

22 Aug 2022

Design and evaluation of an MRI-ready, self-propelled needle for prostate interventions

PONE-D-22-12859R1

Dear Dr. Bloemberg,

We’re pleased to inform you that your manuscript has been judged scientifically suitable for publication and will be formally accepted for publication once it meets all outstanding technical requirements.

Kind regards,

Tommaso Ranzani, PhD

Academic Editor

PLOS ONE

Additional Editor Comments (optional):

Reviewers' comments:

Reviewer's Responses to Questions

**Comments to the Author**

1. If the authors have adequately addressed your comments raised in a previous round of review and you feel that this manuscript is now acceptable for publication, you may indicate that here to bypass the “Comments to the Author” section, enter your conflict of interest statement in the “Confidential to Editor” section, and submit your "Accept" recommendation.

Reviewer #2: All comments have been addressed

2. Is the manuscript technically sound, and do the data support the conclusions?

Reviewer #2: Yes

3. Has the statistical analysis been performed appropriately and rigorously? 

Reviewer #2: Yes

4. Have the authors made all data underlying the findings in their manuscript fully available?

Reviewer #2: Yes

5. Is the manuscript presented in an intelligible fashion and written in standard English?

Reviewer #2: Yes

6. Review Comments to the Author

Reviewer #2: The authors have sufficiently addressed my prior comments and the paper is now ready for publication in my estimation.

7. PLOS authors have the option to publish the peer review history of their article (what does this mean?). If published, this will include your full peer review and any attached files.

Reviewer #2: No

---

## [Editor Report · Acceptance letter]

29 Aug 2022

PONE-D-22-12859R1 

Design and evaluation of an MRI-ready, self-propelled needle for prostate interventions 

Dear Dr. Bloemberg:

I'm pleased to inform you that your manuscript has been deemed suitable for publication in PLOS ONE. Congratulations! Your manuscript is now with our production department. 

Kind regards, 

on behalf of

Dr. Tommaso Ranzani 

Academic Editor

PLOS ONE